# Evaluating the impact of model complexity on flood wave propagation and inundation extent with a hydrologic-hydrodynamic model coupling framework

Jannis M. Hoch[1,2,6,7] *, Dirk Eilander[2,3] *, Hiroaki Ikeuchi[3,4] *, Fedor Baart[2], Hessel C. Winsemius[2,5]

[1]Department of Physical Geography, Utrecht University, P.O. Box 80115, 3508 TC Utrecht, Netherlands
[2]Deltares, P.O. Box 177, 2600 MH Delft, Netherlands
[3]Institute for Environmental Studies, VU Amsterdam, 1081 HV Amsterdam, Netherlands
[4]Department of Civil Engineering, University of Tokyo, Tokyo, 153-8505, Japan
[5]Department of Civil Engineering, TU Delft, 2628 CN Delft, Netherlands
[6]IGDORE Institute, Utrecht, Netherlands
[7]Witteveen+Bos, 7411 TJ Deventer, Netherlands

*Equal author contribution

*Correspondence to*: Jannis M. Hoch (j.m.hoch@uu.nl)

**Abstract.** Fluvial flood events are a major threat to people and infrastructure. Typically, flood hazard is driven by hydrologic or river routing and floodplain flow processes. Since they are often simulated by different models, coupling these models may be a viable way to increase the integration of different physical drivers of simulated inundation estimates. To facilitate coupling different models and integrating across flood hazard processes, we here present GLOFRIM 2.0, a globally applicable framework for integrated hydrologic-hydrodynamic modelling. We then tested the hypothesis that smart model coupling can advance inundation modelling in the Amazon and Ganges basins. By means of GLOFRIM, we coupled the global hydrologic model PCR-GLOBWB with the hydrodynamic models CaMa-Flood and LISFLOOD-FP. Results show that replacing the kinematic wave approximation of the hydrologic model with the local inertia equation of CaMa-Flood greatly enhances accuracy of peak discharge simulations as expressed by an increase of the Nash-Sutcliffe Efficiency (NSE) from 0.48 to 0.71. Flood maps obtained with LISFLOOD-FP improved representation of observed flood extent (critical success index C=0.46), compared to downscaled products of PCR-GLOBWB and CaMa-Flood (C=0.30 and C=0.25, respectively). Results confirm that model coupling can indeed be a viable way forward towards more integrated flood simulations. However, results also suggest that the accuracy of coupled models still largely depends on the model forcing. Hence, further efforts must be undertaken to improve the magnitude and timing of simulated runoff. Besides, flood risk is, particularly in delta areas, driven by coastal processes. A more holistic representation of flood processes in delta areas, for example by incorporating a tide and surge model, must therefore be a next development step of GLOFRIM, making even more physically-robust estimates possible for adequate flood risk management practices.

## 1 Introduction

Globally, the number of exposed population and assets as well as casualties and economic damage due to flooding increased greatly in recent decades (Hirabayashi et al., 2013; Jongman et al., 2012; Ward et al., 2013; Winsemius et al., 2016) . To better predict and understand current and future flood hazard as well as to plan mitigation and adaption measures, several numerical

models, so-called global flood models (Trigg et al., 2016; Ward et al., 2015) were developed to provide current and future estimates.

Current global flood models, however, are tailor-made for certain applications and excel at, for instance, their representation of hydrologic processes, computationally efficient routing, or hydrodynamic surface flow processes. Depending on model structure and workflow, each model has therefore specific advantages and shortcomings. Also, there are marked differences

between the spatial resolutions, affecting both the range of physical processes to be simulated and the applicability of model output maps (Beven et al., 2015; Bierkens et al., 2015).

Additionally, different physical processes may be governing at different spatial scales. For instance, 1D hydrodynamics may be appropriate for large-scale or even global applications, explicitly modelling floodplain flow with 1D/2D models can be vital for more local assessments. Depending on the envisaged application, modelling set-ups must thus be able to reflect the

importance of various flood triggers by integrating across the relevant physical processes and spatial scales. Answering the question of how much complexity is needed can have benefits in avoiding not only under-fitting, but also over-fitting of the problem (Neal et al., 2012b). For instance, applying higher-order approximations of the shallow water equations may be disproportionate for high-gradient regions where channel flow is the main physical process to consider while it is very much needed if inundation patterns in flat delta areas are simulated.

For simulating physical processes and hazards across spatial scales without adding just another new model, flexible computational frameworks are viable means as they can be designed depending on envisaged application. One example is the 'plug-and-play' model coupling tool pyMT ("Python Modeling Tool"; https://csdms.colorado.edu/wiki/PyMT) developed by the CSDMS ("Community Surface Dynamics Modelling System"; Syvitski et al., 2014) which, however, focusses on the whole range of Earth-surface models. By providing the flexibility to couple models depending on the application, fit-for-purpose

coupled models can be created. For instance, one can address different processes that govern at different spatial (and temporal) resolutions by nesting local high-resolution 2D models in large scale 1D models only where these processes are relevant. This is in contrast to other approaches aiming at combining floodplain runoff with river channel routing via pre-defined lateral inflows (Biancamaria et al., 2009; Felder et al., 2017; Lian et al., 2007).

To our knowledge, the development and application of flexible model coupling frameworks specifically designed for large-

scale coupled hydrologic and hydrodynamic modelling is very limited. For example, GLOFRIM, a framework for integrated hydrologic-hydrodynamic modelling, was developed and applied recently (Hoch et al., 2017b, 2018). Both studies coupled the coarse-resolution global hydrologic model PCR-GLOBWB (Sutanudjaja et al., 2018) with the fine-resolution hydrodynamic models Delft3D Flexible Mesh (Kernkamp et al., 2011) and LISFLOOD-FP (Bates et al., 2010) set up for a fraction of the

studied basin only. These studies showed that coupling hydrologic processes with more advanced hydrodynamic processes improves both representation of inundation along reaches as well as the simulation of flood wave propagation.

As the coupling framework was, however, still limited to large-scale hydrologic models and local 1D/2D hydrodynamic models covering the floodplains, flood-triggering processes outside the domain of the hydrodynamic models might be hampered
because of the simplified routing still executed by the hydrologic models.

Adding a river routing component to the model coupling framework allows for potentially improved flood wave propagation throughout the entire domain (Zhao et al., 2017) and makes it possible to focus the computationally heavy 2D hydrodynamics on even smaller domains. Consequently, it would be possible to create various coupled models with different levels of complexity depending which model and model types are combined for which fraction of the study area.

To assess whether and under which circumstances model coupling is beneficial for yielding improved discharge and inundation extent and how additional layers of complexity may benefit output accuracy, we tested different coupling designs of different complexity. Hence, the overarching research objectives of this study are to gain insights in the opportunities as well as challenges of a) establishing a modular and flexible model coupling framework and b) applying coupling configurations of different complexity to two case studies.

To this end, GLOFRIM was evolved by creating a more modular framework, extending the models contained, providing a plug-and-play tool allowing for spatially explicit coupling of hydrologic and hydrodynamic models. To enhance process and scale integration, we added the global river routing model CaMa-Flood (Yamazaki et al., 2011) to improve runoff routing over entire catchments. Besides, we added the wflow hydrologic modelling platform (Schellekens et al., 2018) to give the end-user a greater choice which hydrologic model to use and at which spatial resolution.

With its revised concept, we envisage two applications as core of the new GLOFRIM 2.0  framework: A) fast routing of runoff over large domains and B) detailed local inundation modelling for smaller "spatially nested" areas such as river deltas. Besides, GLOFRIM can also be applied for benchmarking hydrologic and hydrodynamic models.

In the remainder of this article, we will first describe GLOFRIM and the models contained briefly. The benefit of applying a flexible model coupling framework for large-scale inundation modelling is then tested by two applications in the Amazon and
Ganges-Brahmaputra basins, respectively. We conclude with recommendations and an outlook for future applications in integrated flood hazard modelling and assessment.

## 2 The coupling framework and its component models

### 2.1 GLOFRIM 2.0

GLOFRIM 2.0 continues and extends the online and spatially explicit model coupling approach of the previously published
framework GLOFRIM 1.0 (Hoch et al., 2017b). With GLOFRIM 1.0 it was possible to couple the global hydrologic model PCR-GLOBWB (Sutanudjaja et al., 2018) with the hydrodynamic models Delft3D Flexible Mesh (Kernkamp et al., 2011) and LISFLOOD-FP (Bates et al., 2010) by employing the Basic Model Interface (BMI; Peckham et al., 2013). In the new

GLOFRIM version, the models CaMa-Flood (CMF; Yamazaki et al., 2011) and wflow (WFL; Schellekens et al., 2018) have been added. In its current version, GLOFRIM has been tested for one-way coupling only, that is, model output can be exchanged from one model to another but not interchangeably. While this is entirely sufficient for the presented showcases documenting the advances of GLOFRIM, the full potential of online-coupling will only be tapped if also two-way coupling is

fully supported

For further information on the motivation, design, and technical implementation of the BMI itself and within GLOFRIM, we refer to the mentioned articles as well as the online documentations of the BMI (https://csdms.colorado.edu/wiki/BMI_Description) and GLOFRIM (https://glofrim.readthedocs.io).

We decided to employ the BMI concept since it is non-invasive, avoiding entanglement of code from different models. Besides,
the coupling workflow and design can be designed flexibly and altered easily. Flexible coupling via interfaces allows for other models to be added to GLOFRIM once the BMI is implemented and homogenized with CSDSM BMI standards (https://bmi-spec.readthedocs.io/en/latest/). The BMI standard was expanded by adding a step to the initialization of the models. In this step, prior to the actual model initialization, only the configuration is initialized which allows for e,g. changing parameters of the individual models. Furthermore, a common time definition between all models was adopted. In addition to the BMI, a
submodule was added which interprets the model grid type and spatial domain. Using this sub-module, a spatial index of the 2D domain (and 1D network if present) is constructed which allows for straight forward spatial explicit coupling of models.

## 2.2 The supported models

Hereafter, the three models used for the two test cases are briefly outlined. For a complete overview of all five available models currently included in GLOFRIM as well as for a more detailed description of the models, we refer to the supplement
"GLOFRIM 2.0 and description of supported models", the GLOFRIM online documentation, and the model-specific description papers.

### 2.2.1 PCR-GLOBWB

PCR-GLOBWB (PCR; Sutanudjaja et al., 2018) is a global hydrologic model solving the water balance for the entire global terrestrial surface at a daily time step. It is forced with meteorological data such as precipitation, evaporation, and temperature,
which drive hydrologic processes in two vertically-stacked soil layers as well as a bucket-type groundwater module. For all applications, we employed PCR at 30 arc-min spatial resolution which is equivalent to around 50 km x 50 km at the Equator. By default, resulting surface runoff can be routed solving the kinematic wave approximation. To allow for a fair comparison and at least minimum simplistic interaction between channel and floodplain volumes, the "DynRout" extension of PCR was used.

### 2.2.2 CaMa-Flood

CaMa-Flood (CMF; (Yamazaki et al., 2011) simulates the floodplain hydrodynamics of continental-scale rivers globally employing an adaptive time stepping scheme. Since it solves the 1D local inertial equation (Bates et al., 2010; Yamazaki et al., 2013) and only changes in water storage are prognosticated, simulations are computationally efficient. Another advantage

is that CMF is a global model and therefore model data exists for the entire terrestrial surface, reducing the need to manually set up the model. Yet, also this possibility is provided in case more accurate local data is available. Output is provided at 0.25 degree spatial resolution, but inundation depth can be downscaled to 0.005 degree for more accurate assessments.

### 2.2.3 LISFLOOD-FP

LISFLOOD-FP (LFP; Bates et al., 2010) solves the local inertia equations for both channels and floodplains using a sub-grid

channel scheme (Neal et al., 2012a) and adaptive time stepping, allowing for simulating flow not only in longitudinal but also lateral direction. By explicitly simulating floodplain flow, inundation dynamics such as velocity and duration as well as channel-floodplain interactions such as return flows can be captured.

LFP can be discretized at any spatial resolution but is typically employed for fine-resolution assessment of inundation dynamics. The required input data can be produced using common GIS programmes and do not require extensive pre-

15 processing.

### 2.3 Possible coupling realizations

We envision GLOFRIM as a plug-and-play tool where the user can design the coupled model depending on required model complexity. With the increased number of models contained by GLOFRIM 2.0, the number of possible combinations increased

too, see Figure 1. We define three categories of models: i) hydrologic models computing runoff from meteorological data (PCR and WFL; purple in Figure 1); ii) routing models focussing on simulating water transport simulation along a 1D river network (CMF; yellow in Figure 1); and iii) hydrodynamic models determining discharge for both 1D river network and 2D floodplain areas (DFM and LFP; red in Figure 1).

### 3 Model runs

While combination 1) in Figure 1 was already applied and assessed in previous work (cf. Hoch et al., 2017b, 2017a), we merely focus on possibilities 2) and 3) in this study. To test the combinations, we designed two separate test cases to achieve the research objectives: while in test A we assess the opportunities and challenges for advancing the simulation of flood wave propagation by coupling large-scale hydrology with a routing model, test B aims at investigating the benefit of nesting a high-resolution 1D/2D hydrodynamic model into large-scale models for improved local inundation mapping.

### 3.1 Test A: Improving large-scale routing and benchmarking hydrology

#### 3.1.1 Model set-up

One possible application of GLOFRIM is routing hydrologic output over large distances, using more sophisticated flow solvers than implemented in global hydrologic models. To assess the value added, the routing scheme of CMF replaced the kinematic wave approximation as implemented in PCR-DynRout with the local inertia equations in the Amazon River basin. Besides the more advanced solver, the channel network of CMF is resolved at a finer spatial resolution than PCR (Figure 2).

Both CMF and PCR are models with global extent. Consequently, we did not have to create basin specific discretizations but could use the already existing default model set-up after clipping to the needed extent. PCR contains only a Manning's roughness coefficient for channels which was set to 0.03 m$^{-1/3}$ s. The channel roughness coefficient of CMF was aligned with PCR, and the floodplain roughness coefficient was set to 0.10 m$^{-1/3}$ s. No a priori calibration was performed but roughness coefficients selected merely based on previous model applications. For spatially explicit model coupling, simulated surface runoff per PCR cell was provided to CMF using BMI functions and subsequently interpolated within CMF using model-internal routines.

#### 3.1.2 Model validation

We separately ran PCR-DynRout and CMF coupled to PCR (PCR->CMF hereafter) for the period 2007 until 2009, preceded by two years of spin-up for both models. As test basin we focussed on the Amazon River basin as it is characterized by pronounced flood wave propagation and long travel distance. To validate our runs and obtain information on a wide range of time series properties, we used observed discharge from ORE-HYBAM (http://www.ore-hybam.org/index.php/eng) at Obidos and calculated the total Kling-Gupta Efficiency (KGE) and its individual components (linear correlation KGE_r, bias ratio KGE_β, and variability KGE_α) as well as the Nash-Sutcliffe Efficiency (NSE) for extra emphasis on peak flow simulations. For all timeseries analyses we used the hydroGOF package for R (Zambrano-Bigiarini, 2017). Also, we compared the average time difference between observed and simulated peak discharge (τ) by averaging the annual time gap between observed and simulated peak discharge.

#### 3.1.3 Results and discussion

Results show that, although the identical runoff volumes are routed, obtained discharge estimates vary greatly, particularly with respect to the timing of peak discharge (Table 1) and the hydrograph smoothness (Figure 3). In general, both set-ups have skill as expressed by both KGEs being greater than 0.7. While differences in total KGE are marginal, PCR-DynRout only shows better performance when assessing KGE_α indicating that simulated variability represents observations better. PCR->CMF, in turn, is less biased (it is in fact not biased at all as indicated by a KGE_β of 1) compared to PCR-DynRout. The difference in KGE_β, indicating a difference in simulated flood volume of 6 % between the models, is to some extent caused by the coarse model resolution of PCR which results in an overestimation of 1 % of the catchment area compared to CMF

which determines the catchment boundary at a sub-grid level. As a result, a greater volume is routed with PCR-DynRout than with the coupled CMF model. Quantifying the effect of other factors such as the role of evaporation and groundwater infiltration as simulated by PCR-DynRout but not PCR->CMF was outside the scope of this study, yet previous work indicates that these hydrologic processes may impact both discharge volume and flood extent (Hoch et al., 2018).

Also, the correlation KGE_r of PCR->CMF with observation is better than with PCR-DynRout, most likely due to the ruggedness of the hydrograph simulated by PCR-DynRout. This ruggedness, we suspect, stems from the faster response of the kinematic wave approximation in combination with the derived river slopes with of PCR-DynRout compared to the local inertia approximation of CMF. Besides, the first-order routing and volume distribution scheme of PCR-DynRout may have impacted model results.

It is particularly the difference in NSE that is of interest for flood hazard modelling as this measure is most sensitive towards bias of peak discharge. Here, we see that the coupled model PCR->CMF greatly outperforms PCR-DynRout. This finding is in line with previous research showing that the kinematic wave approximation applied by most GHMs ("Global Hydrologic Models") is not suitable for peak flow simulations (Hoch et al., 2017b, 2017a; Yamazaki et al., 2011; Zhao et al., 2017). Particularly for catchments with low gradients such as the Amazon River basin, the absence of local acceleration and advection

terms results in lower peak discharge accuracy.

Related to this finding is that a marked difference can be found in the average time difference between simulated and observed peak discharge. While PCR-DynRout, on average, predicts peak discharge more than two months earlier than observed, adding the CMF routing reduces this to four days. As it is particularly the correct timing of peak discharge, which is important for operational flood forecasting, results show that the higher model complexity of CMF with respect to river routing may be

beneficial for more actionable model results.

### 3.2 Test B: The benefit of local floodplain hydrodynamics

Employing explicit floodplain flow solvers may be particularly essential for low-lying and flat delta areas, but not throughout the entire basin as run times would increase greatly. Thus, GLOFRIM allows for spatially nested modelling; that is, the local hydrodynamic model is embedded in the basin-wide hydrologic and routing models.

### 3.2.1 Model set-up

In test B, we assess the impact of adding hydrodynamic models for both the river routing as well as floodplain inundation processes. We compared three model coupling configurations with increasing complexity: PCR-DynRout, PCR->CMF (see configuration 2 in Figure 1), and PCR->CMF->LFP (see configuration 3 in Figure 1) for the Ganges-Brahmaputra basin. While PCR still provides the runoff forcing for CMF for the entire catchment, various boundary conditions apply for the

hydrodynamic model; that is, upstream discharge from CMF, local runoff from CMF, and downstream water level dynamics as prescribed within LFP itself (in this case 0 m).

Similar to test A, we used the global default model set-ups of PCR and CMF and clipped them to the extent of the Gangs-Brahmaputra basin. What is different to test A is, however, that we had to perform a calibration of CMF floodplain roughness coefficients and channel depth due to initially insufficient accuracy of discharge by PCR->CMF. Therefore, we applied a Manning's coefficient of 0.03 m$^{-1/3}$ s for PCR as well as for both river and floodplains in CMF. Channel depth was also increased by changing the first factor from its default value 0.14 to 0.20 in the subsequent equation:

$$B = \max[0.14 R_{up}^{0.40}, 2.00]$$

where $B$ is channel depth [m] and $R_{up}$ is the annual maximum of 30-day moving average of upstream runoff [m$^3$ s$^{-1}$].

For the LFP model, we made use of the underlying surface elevation and channel dimension raster data of CMF at 18 arc-seconds and created a LFP discretization for a small domain at the river delta at identical spatial resolution (Figure 4).

### 3.2.2 Model validation

To assess the quality of discharge simulations, we calculated the KGE and its components as well as the NSE based on simulated and observed values for two locations: Hardinge Bridge in the Ganges river and Bahadurabad in the Brahmaputra river (see Figure 4). Observed values were kindly provided by the Institute of Water Modeling, Bangladesh, and the Bangladesh Water Development Board. As both locations lie outside the LFP domain, we could only validate the PCR-DynRout and PCR->CMF at those two locations and therefore had to perform a separate analysis of the PCR->CMF->LFP run. For this, we qualitatively compared the model results from all three model settings at a (arbitrarily chosen) location close to the river mouth (see Figure 4), setting simulated discharge from all three set-ups into relation. In contrast to test A, we desisted from determining τ as there is more than one peak per flood season which complicates an unambiguous analysis.

Additionally, the simulated inundation maps were compared with observed imagery. Therefore, PCR and CMF maps were first downscaled to a resolution of 1 km and 500 m, respectively, making use of their model-specific downscaling routines. As validation data, 8-day composite MODIS imagery of 2007 was used (see Kotera et al., 2016) as this year was characterized by strong monsoon-induced inundations (Islam et al., 2010).

We compared simulated results at date 2007-08-18, the day of maximum total flood extent in the CMF model, of all models with the corresponding 8-day composite MODIS image. The motivation for this approach was threefold: first, to not use LFP output as its output validation should be unaffected by previous decisions; second, because downscaled CMF output has higher resolution than PCR; third, to be able to assess differences in both timing and magnitude of simulated inundation extent. To guarantee comparability, maps of both observed and simulated flood extent were clipped to the LFP model domain and resampled to 500 m spatial resolution applying the nearest neighbour approach.

Inundation extent was validated for all set-ups following the approach of Fewtrell et al. (2008). Thereby, the hit rate H, the false alarm ratio F, and the critical success index C were determined for each inundation map with respect to observed MODIS extent. H, F, and C were computed with the subsequent equations where $N_{obs}$ and $N_{sim}$ indicate the number of inundated cells according to observations and the simulation result under consideration, respectively.

$$H = \frac{N_{sim} \cap N_{obs}}{N_{obs}}$$

$$F = \frac{N_{sim} \backslash N_{obs}}{N_{sim} \cap N_{obs} + N_{sim} \backslash N_{obs}}$$

$$C = \frac{N_{sim} \cap N_{obs}}{N_{sim} \cup N_{obs}}$$

All parameters can vary between 0 and 1. While H=1 shows that all inundated cells in the benchmark data are also inundated in the comparison data, F=1 indicates that the inundated cells in the comparison are entirely false alarms with respect to the benchmark. The critical success rate C, in turn, should be 1 for perfect agreement, thereby penalizing for both under- and overestimation.

### 3.2.3 Results and discussion

**Simulated discharge**

Validating discharge simulated by PCR-DynRout and PCR->CMF at Hardinge Bridge (Ganges river) and Bahadurabad (Brahmaputra river) shows slightly opposite behaviour than the previous test A. For both the Ganges and the Brahmaputra, PCR unexpectedly outperforms the coupled set-up, with results generally being more accurate for the Ganges river (Figure 3, Table 2).

These results show that added models with higher complexity does not always yield actually better results, as in this case the local inertia equations solved by CMF did not outperform the kinematic wave approximation of PCR. The local inertial equation is derived by neglecting only the advection term in the shallow water equations as advection is insignificant for many natural river and floodplain flow conditions with low gradients (de Almeida and Bates, 2013; Hunter et al., 2007; Yamazaki et al., 2013). The kinematic wave approximation, however, only accounts for channel and friction slope. While the reduced physics are less impacting for high-gradient areas, such as mountainous areas, or areas with clearly incised river channels, the Ganges-Brahmaputra basin is characterized by its large and flat floodplains. From a theoretical point of view, models applying the local inertia equations should therefore outperform simpler routing schemes in this study area. Yet, this is not the case and thus points to one of the key structural challenges with such cascading one-directional model coupling: while the most advanced hydrodynamic schemes can be added, the overall model accuracy still depends greatly on model data and parameter uncertainties, calibration, and both the meteorological and hydrologic forcing. Recent research showed, for instance, that the meteorological data set used can be a key control of discharge accuracy (Towner et al., 2019). Note also that PCR-DynRout and CMF use different topography and river bathymetry data as well as different river network concepts (i.e. flexible location of waterways (FLOW; Yamazaki et al., 2009) in CMF compared to eight directions toward neighbouring cells (D8) in PCR DynRout) to derive the routing schematization. The differences in modelled discharge can therefore not only be attributed to the difference in approximation of the shallow water equations.

Benchmarking simulated discharge from PCR, PCR->CMF, and PCR->CMF->LFP corroborates that including 2D floodplain flow processes reduced the volume routed along the main river channel whereas the timing of peak discharge does not deviate markedly between the coupled set-ups (Figure 6). In combination with the simulated flood extent of PCR->CMF->LFP (Figure 7d), the difference in discharge rate in the main channel can be attributed to the additional flow through smaller side channel which is only possible if 2D flow is explicitly modelled. Even though no validation is possible due to the lack of observed data within the LFP domain, the already prevailing underestimation of discharge by PCR->CMF let's one speculate that PCR->CMF->LFP is less accurate in resembling discharge magnitude. Since CMF and LFP use the same underlying data for deriving river geometry and floodplain topography, we assume that model internal, input data independent, factors result in the deviations between the CMF and LFP. At the current stage, unfortunately, an unambiguous explanation cannot be made yet as a further investigation exceeds the scope of this study.

**Inundation extent**

Simulating inundation maps can benefit greatly from adding 2D hydrodynamic floodplain flow computations. Validating the downscaled inundation maps from PCR and PCR->CMF with the modelled results of PCR->CMF->LFP shows significant deviations between model set-ups (Figure 7). In fact, results insinuate that acceptable representation of inundation patterns as expressed by the critical success index C can only be achieved by also accounting for floodplain flow and discharge through side channels (Table 3).

Table 3 furthermore shows that, despite having comparable false alarm ratios, the hit rate H is much higher for PCR->CMF->LFP and, in turn, so is the critical success index C. The differences in hit rate are largely resulting from simulated inundations along smaller water bodies, especially compared to PCR->CMF, and by simulating the extent across the entire river floodplain, which is particularly not the case for PCR (Figure 7). It is for those areas, which may not necessarily directly adjacent to the main river stem, that downscaling procedures based on volume or water depth distribution curves may not suffice to represent the actual locally-relevant flood-triggering processes, leading to a low hit rate.

It is interesting to see that PCR->CMF does not show any inundation for a part of the main river reach of the Ganges-Brahmaputra. This is the result of the combination of the unit catchment scheme used in CMF where different river reaches may have different geometry and the static downscaling approach. The deviations to PCR->CMF->LFP are hence a function of model internal specifications as the underlying input data is identical. Consequently, this comparison hints at positive impact of models with higher complexity explicitly modelling of floodplain flow instead of downscaling, and that the threshold for LFP to predict excess channel volume may be lower than for CMF.

Despite all efforts to make the validation as fair as possible, there are still some limitations that must be kept in mind. For example, inundation patterns of the Ganges-Brahmaputra delta are largely affected by tide and surge dynamics (Ikeuchi et al., 2015). Since we discretized all models with a steady 0 m water level boundary, it must be acknowledged that a perfect fit between observations and simulations would not be possible. Besides, the downscaling routines of PCR and PCR->CMF employ different approaches and data, resulting in locally marked differences in results. Aligning the routines was, however, outside of the scope of this study. The arising issues of different inundation extent due to different model routines and data

was already discussed by other studies and remains subject to on-going debate how to minimise the gap between models (Bernhofen et al., 2018; Hoch and Trigg, 2019; Trigg et al., 2016). Last, it is important to state that no calibration of the models with respect to simulated flood extent was performed. While this gives a fair picture of what a model is capable of under genuine conditions, the values here presented do not reflect that actual potential of each model.

## 5  4 Conclusions, recommendations, and outlook

We developed GLOFRIM 2.0, a globally applicable framework for integrated hydrologic-hydrodynamic modelling, to evaluate the added value of model coupling and applying models with varying complexity for discharge and flood extent simulations, testing it in two case studies. By combining hydrology and hydrodynamics in a 'plug-and-play' way, it is possible to integrate across a suite of flood hazard drivers and design different coupled models, each having another level of varying complexity
while maintaining identical spatially-varying model forcing.

In this context, the main conclusions are:

- For discharge simulations, applying models with higher complexity is beneficial. By replacing the kinematic wave approximation with the local inertia equations, obtained results can be improved given the model schematization and runoff forcing itself are accurate. Including more complex 1D/2D hydrodynamic processes does not further improve
discharge simulations compared to 1D simulations.
- For inundation extent simulations, employing a model capable of explicitly simulating floodplain flow and channel-floodplain interactions at a fine spatial resolution outperforms less complex models, particularly those using a downscaling approach. Using a 1D/2D model can be of added value for those areas where no river network is present.

Results therefore suggest that including additional layers of complexity can indeed benefit model accuracy, yet this depends
on the output variable under consideration. This means that for some applications opting for less complex model compositions suffices to obtain accurate results, possibly even at shorter run times.

Besides, the findings made lead to several important conclusions concerning wider implications of coupling different models and model components:

- (Re-)calibration of coupled model may be needed when replacing a native process from one model, for instance the
kinematic wave routing in PCR, with the same process in another model, for instance routing based on the local inertia equations. While we did calibrate the Manning coefficient and channel depth parameters of CMF in the present study, new strategies for the calibration of coupled models might be required before yielding improved results.
- Before full use can be made of adding models with higher complexity, it should be ensured that the model forcing (here runoff) is accurate. Wrongly timed runoff routed with simplistic routing schemes can still yield right results for
the wrong reasons, runoff simulated by hydrologic models needs to be validated before employing in a modelling cascade.

Our evaluation furthermore shows that including floodplain flow and discharge through secondary channels is paramount for accurately simulated inundation maps. Notwithstanding the best performance of this set-up, a critical success index of 0.46 indicates that only around half of the actual extent is correctly captured by the model, leaving much room for improvement.

For example, a thorough analysis of the used DEM may help to reveal whether water is trapped in local depressions, hampering return flows and therefore increasing the flood extent unnecessarily. Possible problems could be solved by hydraulic conditioning (Yamazaki et al., 2012) or by updating the model data with the newly developed MERIT-DEM (Yamazaki et al., 2017). It is hence worth noting that in this study, we did not optimize the schematizations of the models involved for the sake

of an untarnished evaluation of the impact of their complexity. Yet, GLOFRIM 2.0 can also be applied for more bespoke studies where optimized schematizations may be used, and thus overall evaluation scores will most likely increase strongly. Furthermore, as the river networks for CMF and PCR DynRout are setup following a different conceptualization, it was not possible to eliminate the effect of river schematization from the shallow water equation approximation.

Despite the progress made in integrating various drivers of flood hazard, this is still limited to fluvial processes. Despite the

10 predicted increase in fluvial flood hazard, most low-lying delta regions are under even greater threat from coastal flood events (Tessler et al., 2015; Visser et al., 2012). Additionally, compound events of simultaneous high discharge and high sea level will have to be included in future flood hazard estimates in many delta regions (Ikeuchi et al., 2017; Ward et al., 2018). Thus, it will be necessary to integrate fluvial and coastal flood hazard too. One avenue would be to use dynamic sea level boundaries. Another, more advanced, option would be to include surge and tide models into GLOFRIM such as the Global Tide and Surge

Reanalysis product (Muis et al., 2016) once they are equipped with a Basic Model Interface. In addition to increasing the number of representable processes, future work will also focus on testing and further developing of a two-way coupling scheme which would allow for more complex integrations and mutual update of prognostic variables between models. By linking hydrologic, routing, and hydrodynamic models, we can establish a model cascade which can simulate the inundation-driving processes from the mountains to the coast. As such, GLOFRIM 2.0 can be a key tool for future more holistic modelling studies

researching the effect of the interplay of meteorology and hydrology, river routing, and floodplain dynamics on flood hazard and risk. Being designed as a plug-and-play tool, flexible coupling frameworks could thus provide scientific evidence supporting decision-making and risk management for a wide range of conditions.

**Data availability**

GLOFRIM 2.0 code is stored online and free to use, spread, and modify under the GNU GPL 3.0 license at https://doi.org/10.5281/zenodo.597107. The model input data as well as schematizations and scripts used for the analysis can be found at https://doi.org/10.5281/zenodo.3346803 under MIT license. For further information regarding GLOFRIM v2.0 and the currently supported models, we refer to the relevant papers as well as to the supplement.

**Author contributions**

We would like to acknowledge that this work has profited from equal contributions of JMH., DE, and HI. Thus, JMH, DE, and HI were all responsible for re-structuring and adding of code, performing test runs, and validating model results. FB and HI implemented the BMI-adapters into CaMa-Flood while FB and JMH implemented it into LISFLOOD-FP. JMH and DE designed the updated workflow of GLOFRIM 2.0. All authors contributed to the manuscript.

**Acknowledgments**

We want to thank Willem van Verseveld and Mark Hegnauer (both Deltares) for the help with WFLOW as well as Rens van Beek and Edwin Sutanudjaja (both Utrecht University) for the help with PCR-GLOBWB. JMH was financed by the EIT Climate-KIC programme under project title "Global high-resolution database of current and future river flood hazard to support planning, adaption and re-insurance". DE received funding from NWO VIDI grant no. 016.161.324. HI was financially supported by Japan Society for the Promotion of Science (JSPS) KAKENHI grant number JP16J07523. Observed river discharge data for the Ganges and Brahmaputra rivers were kindly provided by the Institute of Water Modeling, Bangladesh, and the Bangladesh Water Development Board.

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

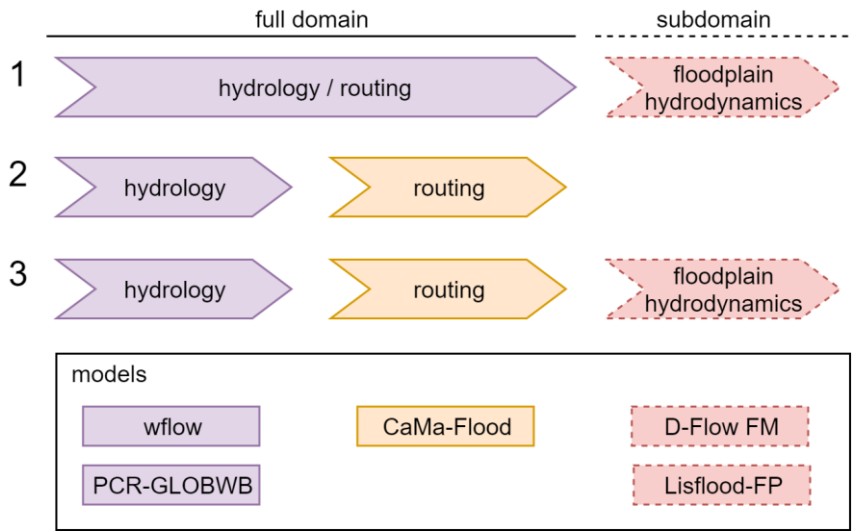

**Figure 1: Supported and tested coupling configuration and supported models of GLOFRIM 2.0. We distinguish between hydrologic (purple), routing (yellow) and floodplain (2D) hydrodynamic models (red).**

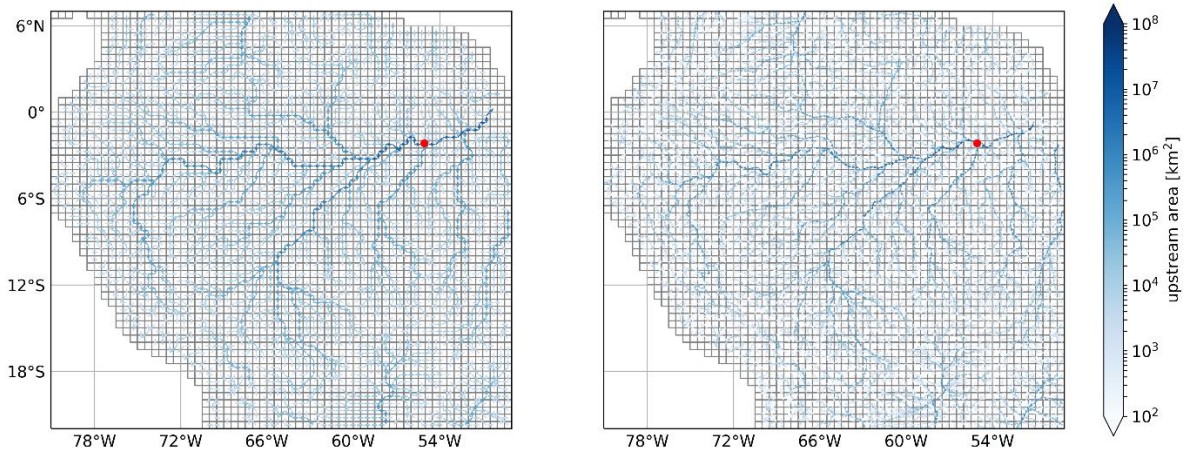

5    **Figure 2: The PCR drainage network (LDD) of the Amazon River basin at 30 arc-min spatial resolution (left) as well the 1-D channel network of CMF (right). The upstream area of drainage network is shown in blues and the location of the Obidos gauge in red.**

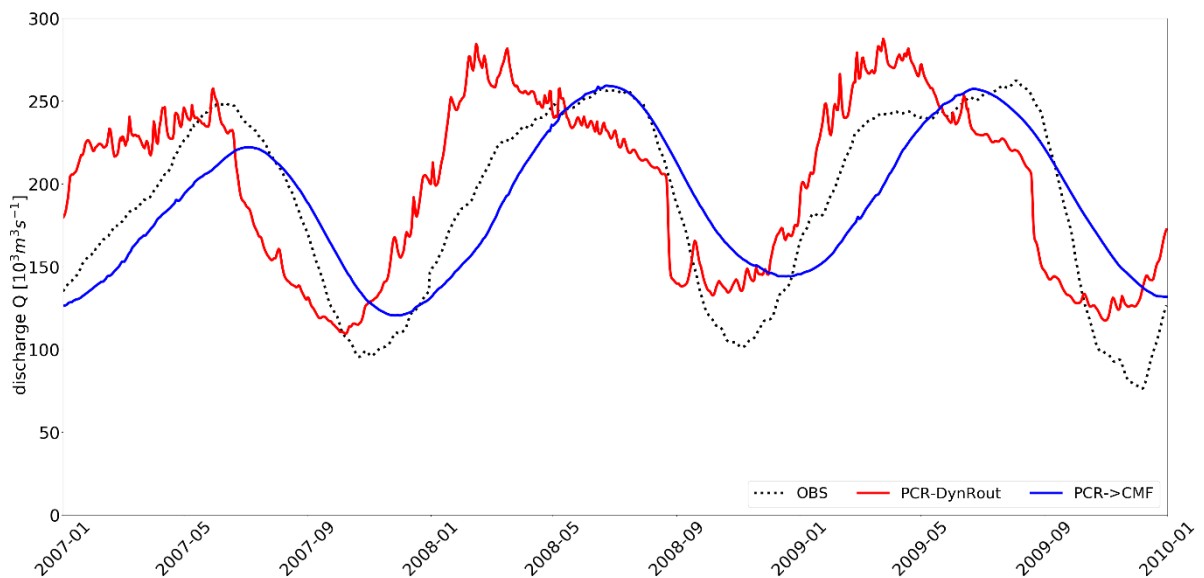

**Figure 3: Simulated discharge by PCR-DynRout and PCR->CMF as well as observed discharge at Obidos**

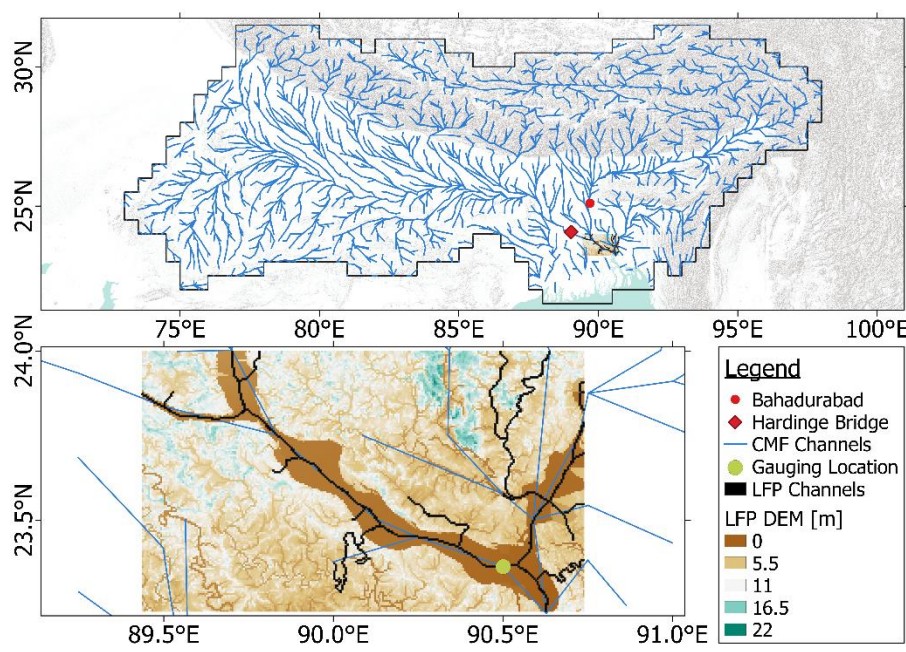

**Figure 4: (top) CMF channel network in Ganges-Brahmaputra basin as well as locations of observation stations Hardinge Bridge and Bahadurabad for validating model output from both PCR and CMF; (bottom) zoom to LFP extent showing LFP DEM and channel network as well as gauging stations where output from PCR, CMF, and LFP is compared**

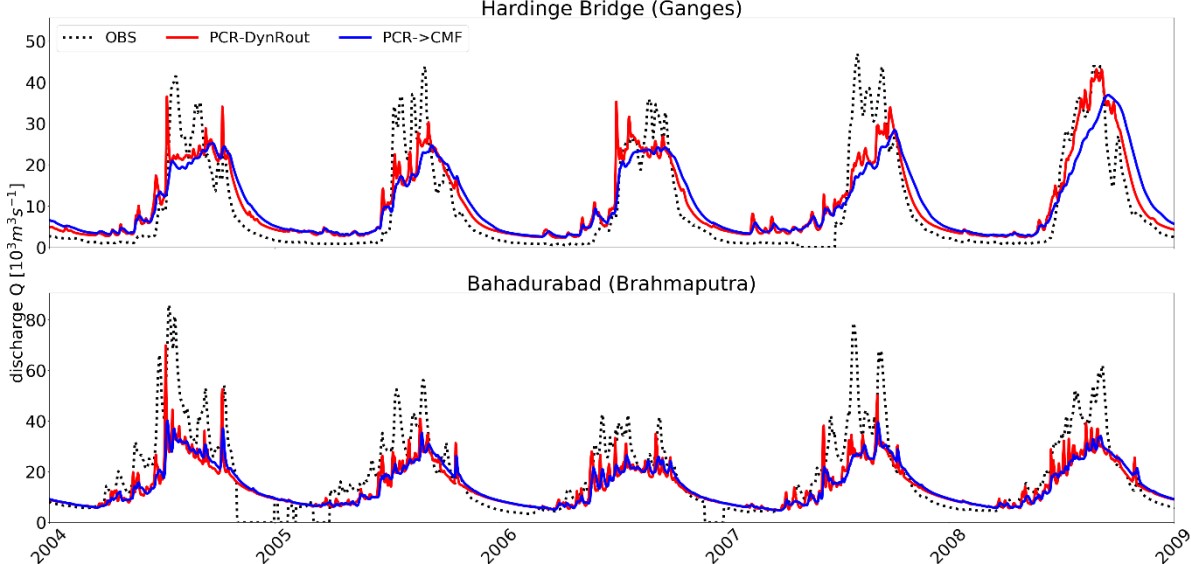

**Figure 5: Simulated discharge by PCR-DynRout and PCR->CMF as well as observed discharge at both Hardinge Bridge (Ganges) and Bahadurabad (Brahmaputra)**

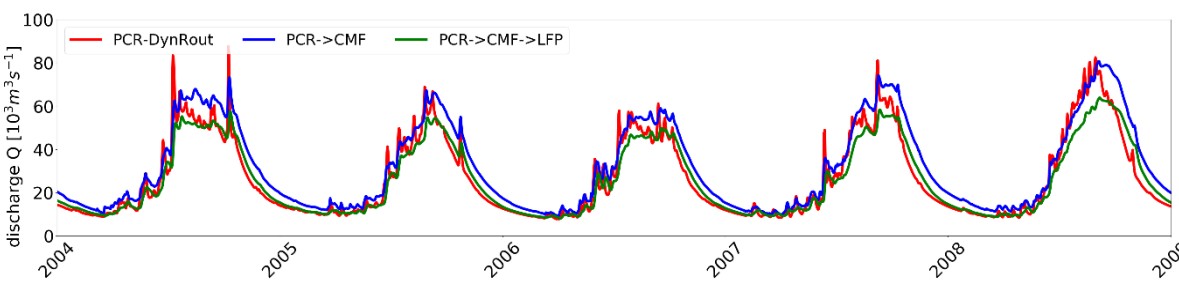

**Figure 6: Simulated discharge from PCR-DynRout, PCR->CMF, and PCR->CMF->LFP at the common observation point as depicted in Figure 4**

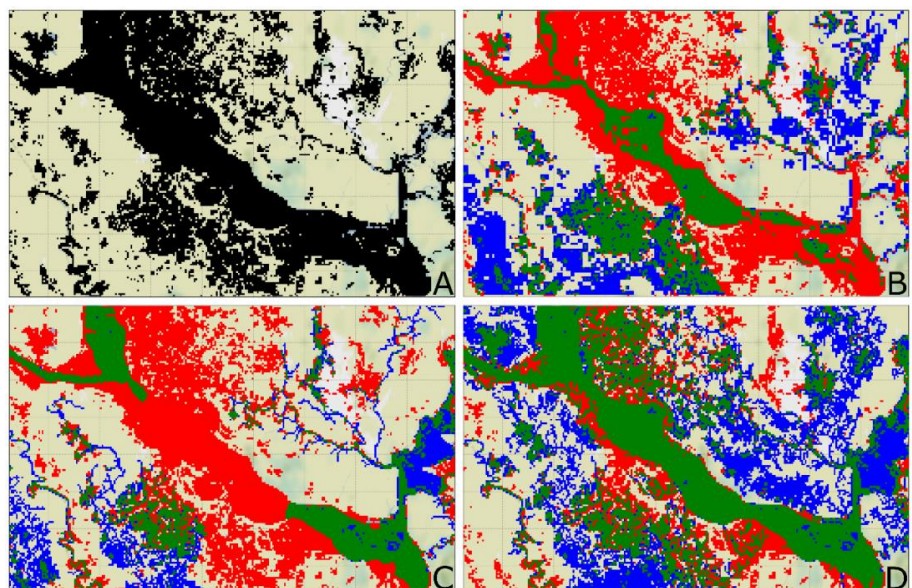

**Figure 7: (A) observed flood extent by MODIS; overlay of observed and modelled flood extent for (B) downscaled PCR, (C) downscaled PCR->CMF, and (D) PCR->CMF->LFP where blue indicates model only, red indicates observation only, and green indicates agreement between model and observations**

5  **Table 1: Assessment of performance of PCR-DynRout and PCR->CMF runs at Obidos; the shaded boxes indicate better performance**

|  | **KGE** | **KGE_r** | **KGE_β** | **KGE_α** | **NSE** | **avg τ** |
|---|---|---|---|---|---|---|
| **PCR-DynRout** | 0.72 | 0.74 | 1.06 | 0.91 | 0.48 | 89 d |
| **PCR->CMF** | 0.71 | 0.85 | 1.00 | 0.75 | 0.71 | 4 d |

**Table 2: Assessment of performance of PCR, PCR->CMF, and PCR->CMF->LFP runs at both Hardinge Bridge (Ganges) and Bahadurabad (Brahmaputra); the coloured boxes indicate best performance compared to other set-ups**

|  | **KGE** | **KGE_r** | **KGE_β** | **KGE_α** | **NSE** |
|---|---|---|---|---|---|
| Hardinge Bridge (Ganges) | | | | | |
| **PCR-DynRout** | 0.71 | 0.89 | 1.15 | 0.78 | 0.77 |
| **PCR->CMF** | 0.63 | 0.83 | 1.15 | 0.70 | 0.66 |
| Bahadurabad (Brahmaputra) | | | | | |
| **PCR-DynRout** | 0.46 | 0.84 | 0.79 | 0.52 | 0.55 |
| **PCR->CMF** | 0.44 | 0.86 | 0.79 | 0.50 | 0.54 |

**Table 3: Hit rate, false alarm ratio, and critical success index for the three model set-ups**

|  | **PCR** | **PCR->CMF** | **PCR->CMF->LFP** |
|---|---|---|---|
| **Hit rate** | 0.38 | 0.30 | 0.70 |
| **False alarm ratio** | 0.44 | 0.40 | 0.42 |
| **Critical success index** | 0.30 | 0.25 | 0.46 |