# Peer review of "Evaluating the impact of model complexity on flood wave propagation and inundation extent with a hydrologic-hydrodynamic model coupling framework"

_Natural Hazards and Earth System Sciences, 2019_

## Referee Comment (RC1) · Guy J.-P. Schumann (Referee) · 6 Apr 2019

This paper looks at the coupling of global hydrologic and hydraulic models to simulate flood hazard in three major river systems.

The paper is generally well written and the topic should be of interest to a wide audience. The coupling of models is definitely an interesting topic.

Several important points need clarification:

Title: Not sure what the word "nested" here implies, please consider removing it. The models are not placed inside another. Rather they are used within a framework as a

one-way coupling.

Abstract: Please only use "are" in the first sentence, three tenses are confusing.

Abstract: The word "physicality" I think is wrong here. Maybe say increase the physics. . .

Introduction: This is an account of GLOFRIM. It would be useful to also give a bit of background literature on other attempts to include floodplain representation in global hydrologic models for instance.

Methods: All three main models are described rather very briefly. More detail is needed here to outline the physics somewhat more clearly. Also, why not use CaMaFlood as a 2D floodplain model – it has a subgrid floodplain representation with simplified local inertia included? What is the added value of LISFLOOD-FP and why use CMF only as a 1D routing here? This needs to be better explained.

Coupling realizations. This may be outside the scope of this paper but the real benefit of any coupling is that the models can communicate in a two-way feedback, in other words, one should ideally be able to use the computed (error) in inundation to adjust or correct inflow to the hydrologic model. I think realizations here are only one-way coupling and this needs to be made clearer. The strength of the GLOFRIM framework is dynamic coupling (but one way) within a plug 'n play model builder.

Model validation: Validation values NSE and hit rate, and CSI are quite low. An explanation for these relatively low values should be included

Discussion/Conclusion: Again, here the limitations of one-way coupling should be more discussed.

In Fig 5, I believe the model simulation with LFP is missing. Please check.

---

## Author Comment (AC1) · 25 Apr 2019

Dear dr. Schumann,

Many thanks for your swift review on the manuscript. We also want to express our thankfulness for the kind words on the submitted article, for raising your concerns about shortcomings of the current version of the manuscript, and for providing constructive criticism how to further improve the work.

Hereafter, we address your remarks made in the review document per point and outline how we will adapt the manuscript to account for all concerns raised. Thereby we first refer to your original comments in black and subsequently provide our response in blue.

Title: Not sure what the word "nested" here implies, please consider removing it. The models are placed not inside each other. Rather they are used within a framework as a one-way coupling.

Many thanks for outlining this ambiguity. From the perspective of model code, it is indeed correct that the models (i.e. model codes) are not placed inside each other but remain individual entities coupled by means of a framework. However, this is not what we are referring to in the title. Since we employ models with different spatial extent, for example one for the entire basin and one for only the delta area, the models are "geographically" placed inside each other and hence nested. As providing this explanation in the title would make it too lengthy we will stick with "nested" in the title but will ensure that it's clearly formulated in the main text.

Abstract: Please only use "are"in the first sentence, three tenses are confusing.

Thank you for this remark. We will reduce the number of tenses to one to make the sentence better digestible and comprehensive.

Abstract: The word "physicality" I think is wrong here. Maybe say increase the physics

Thanks for the comment. After carefully re-reading the abstract we concur with you and will rephrase the sentence accordingly.

Introduction: This is an account of GLOFRIM. It would be useful to also give a bit of background literature on other attempts to include floodplain representation in global hydrologic models for instance.

We thank you for this thoughtful comment. In fact GLOFRIM is not so much designed to include floodplain representation in global hydrologic models, but to facilitate model coupling in general. Nevertheless, we see the added value of extending the discussion on other approaches trying to include floodplain representation in global hydrologic models or similar model (component) integration approaches. Thus, we will increase the literature review on this topic in the revised manuscript.

Methods: All three main models are described rather very briefly. More detail is needed here to outline the physics somewhat more clearly. Also, why not use CaMaFlood as a 2D floodplain model – it has a subgrid floodplain representation with simplified local inertia included? What is the added value of LISFLOOD-FP and why use CMF only as a 1D routing here? This needs to be better explained.

We appreciate this critical comment on model description and choice. Regarding the former, we will carefully re-assess the manuscript as well as the supplement. Where needed, we will extend the model description. As we aim to keep the model description in the actual manuscript as brief as possible, most of the extension will most likely be in the supplement; yet, the current version of the manuscript may be extended as well to ensure the minimum required to follow and comprehend the presented test cases.

Regarding the CaMa-Flood model, floodplain inundation depths at a high resolution are the result of statically downscaled flood volumes contained in the 1D floodplain storage and not the result of 2D dynamical computations (see the model manual here: http://hydro.iis.u-tokyo.ac.jp/~yamadai/cama-flood/Manual_CaMa-Flood_v362.pdf). This is very different to LISFLOOD-FP which dynamically computes inundation depths on a high resolution throughout the floodplain based on the local inertial equations. In that sense, we consider

CaMa-Flood and LISFLOOD-FP to be different categories of models as also shown in the manuscript.

We concur that this categorization and hence also the way how models were applied can be explained better. In the revised manuscript, we will add the required information.

Coupling realizations. This may be outside the scope of this paper but the real benefit of any coupling is that the models can communicate in a two-way feedback, in other words, one should ideally be able to use the computed (error) in inundation to adjust or correct inflow to the hydrologic model. I think realizations here are only one-way coupling and this needs to be made clearer. The strength of the GLOFRIM framework is dynamic coupling (but one way) within a plug 'n play model builder.

Many thanks for the great comment on model coupling and its benefits. We agree that model coupling, especially online coupling, is most useful if applied for two-way coupling as it becomes possible to study the dynamic interactions between various physical processes. Even though current work is ongoing (see eg. AGU 2018 abstract on including MODFLOW https://agu.confex.com/agu/fm18/meetingapp.cgi/Paper/389133https://agu.confex.com/agu/fm18/meetingapp.cgi/Paper/389133), this was not yet applied for this manuscript. To be able to make use of the benefits of two-way coupling, we designed GLOFRIM with online coupling already although it would technically not be needed to combine the models used here. For the revised manuscript, we will point out more explicitly the reasoning and benefits for opting for one-way coupling as well as the shortcomings compared to two-way coupling. That way possible ambiguity will be avoided.

Model validation: Validation values NSE and hit rate, and CSI are quite low. An explanation of these relatively low values should be included.

Thanks for the great comment concerning the results of model validation.

- The NSE is only low for the PCR run, not for PCR-CMF. This is due to the kinematic wave equation which lacks the physical reality required to properly estimate flood wave propagation. Besides, the simplicity of the approximation together with the LDD and raster-based routing scheme of PCR produce strong daily variations in discharge (see plot) which yield low NSE values. The updated manuscript will extend the discussion of those results.
- The hit rate is low for PCR and PCR-CMF due to their static downscaling approach which limits simulated inundation extent to areas directly adjacent to (main) rivers mostly. Smaller inundations in areas where for instance backwater effects of 2D floodplains are dominant, cannot be simulated correctly. Besides, the Ganges-Brahmaputra-Meghna delta is well known for its large amount of small river reaches and consequent flow divergence. While PCR cannot account for river bifurcations at all, CMF could potentially but we did choose not to activate this option to maintain a certain common standard between models. This altogether results then in a low hit rate. The revised manuscript will contain this important discussion.
- Similarly, the CSI is low, partially due to the same line of reasoning. Is is furthermore worth mentioning that we did not perform any calibration of the models with respect to inundation extent. We decided to desist from calibration to not make the calibration technique and data dominant over the actual model set-up and conceptualization. This may not have become clear in the current manuscript and thus a more elaborated explanation of modelling choices will be added to the revision.

Discussion/Conclusion: Again, here the limitations of one-way coupling should be more discussed.

Following from your previous comment, the updated version of the manuscript will include a more elaborate discussion about the one-way coupling approach employed in this study.

At Fig. 5, I believe the model simulation with LFP is missing. Please check.

We thank you for spotting this error. Since this figure represents the validation of simulated discharge outside of PCR and CMF for a point outside the LFP domain, it is not the plot that is wrong but the caption. In the updated version of the manuscript we will rectify this error and ensure figure and caption match.

---

## Referee Comment (RC2) · Anonymous Referee #2 · 8 May 2019

In this article, the authors introduce an updated version of a model coupling framework, then use the framework to assess whether adding complexity through a coupled model system can improve the quality of simulated results; here, in the context of flood hazards, calculations of discharge and flood extent.

I found this paper clear, generally well-written, and interesting. My comments are mostly minor, and mainly serve to help clarify certain points and to make the article more accessible to a reader. My one major concern is that the paper lacks scientific heft. In general, it felt more like a demonstration of GLOFRIM than a scientific inquiry where GLOFRIM was the tool for discovery. However, the techniques and the results

presented contribute to scientific progress, and should be shared with the community.

I recommend this article for publication, with the condition that the authors respond to the major and minor comments I've listed below.

**1  Major comments**

I found the discussion of the model results in Sections 3.1.3 and 3.2.3 to be too descriptive; i.e., they were simply a rehashing of the statistics presented in Tables 1, 2, and 3. Rather than summarizing statistics, I would like to know if there's some greater insight that could be gained from these runs. I would like the authors to put a little more effort into interpreting the results. Even a few sentences would be helpful.

**2  Minor comments**

The wording in the paper is often a bit awkward. It could benefit from one more read-through by the authors.

Suggested corrections are listed below by (page number, line number).

- (1, 21) Define NSE before using acronym

- (1, 28) "physically-robust": Remove hyphen

- (3, 11) Recommendation: cite the work of J. Syvitski in this area. Perhaps "challenges of a) establishing a modular and flexible model coupling framework (e.g., Syvitski et al., 2014) and b) applying..."

- (3, 18) "envision" instead of "envisage"

[Figure]

- (3, 19) "Further" instead of "Besides"

- (3, 25) It's odd to title this section "GLOFRIM 2.0" when the subheadings are of the models that are included within it. A better section title might be "The coupling framework and its component models". Section 2.1 could then be titled "GLOFRIM 2.0".

- (4, 3) BMI docs: https://bmi.readthedocs.io/

- (4, 6) Remove link

- (4, 14) Appendix A is not present

- (4, 17) Some of the authors of this paper are also authors of the PCR-GLOBWB model. I have no correction to offer here, but this just feels a little odd. Perhaps the paper might be stronger if an outside model had been used.

- (5, 15) D-Flow FM is referenced here and in Figure 1, but not discussed in the paper

- (6, 7) What are the time steps of the models?

- (6, 8) "focused" instead of "focussed"

- (6, 10-12) Why are KGE and NSE chosen? One or two sentences on why these are the appropriate measures would be useful.

- (6, 18) Why is KGE > 0.7 significant?

- (6, 19-20) What are $KGE_\alpha$ and $KGE_\beta$? I can guess what they are, but they should be explained.

- (7, 17) Missing end bracket in equation for $B$

- (9, 14) Strike "yet" at the end of the sentence

- (11, 10) I applaud the authors for making their code publically available. In the interest of open science, are the model runs available, as well? ("No" is an acceptable answer here.)

**3  References**

Syvitski, J. P. M., E. W. H. Hutton, M. D. Piper, I. Overeem, A. J. Kettner, and S. D. Peckham (2014), Plug and play component modeling—the CSDMS 2.0 approach, in *Proceedings of the 7th Intl. Congress on Env. Modelling and Software*, edited by R. A. Ames D.P., Quinn N.W.T., International Environmental Modelling and Software Society (iEMSs), San Diego, CA.

---

## Author Comment (AC2) · 29 May 2019

Dear anonymous reviewer,

We kindly thank you for critically reading the submitted manuscript and providing helpful comments and questions on current shortcomings.

Hereafter, we will react to your major comment first. Subsequently, we will go through the minor comments whereby we will implement the grammar and spelling comments in the revised manuscript and thus are not considered here explicitly. For better readability, we first state your comment in black and subsequently add our reaction in blue.

**Major comment**

I found the discussion of the model results in Sections 3.1.3 and 3.2.3 to be too descriptive; i.e., they were simply a rehashing of the statistics presented in Tables 1, 2, and 3. Rather than summarizing statistics, I would like to know if there's some greater insight that could be gained from these runs. I would like the authors to put a little more effort into interpreting the results. Even a few sentences would be helpful.

We thank the reviewer for this constructive criticism. While the output statistics are the foundation of a good discussion, we can follow the reviewer's reasoning. In the revised manuscript, we will have elaborated more extensively on the model results and on their wider implications for inundation modelling and hazard assessments.

**Minor comments**

- Define NSE before using acronym.
  We will provide the necessary definition in the abstract
- Recommendation: cite the work of J. Syvitski in this area. Perhaps "challenges of a) establishing a modular and flexible model coupling framework (e.g., Syvitski et al., 2014) and b) applying...".
  Many thanks for providing this reference  and a way to include in the manuscript. We will add the reference to the revised manuscript at a sensible location in the text.
- It's odd to title this section "GLOFRIM 2.0" when the subheadings are of the models that are included within it.   A better section title might be "The coupling framework and its component models". Section 2.1 could then be titled "GLOFRIM 2.0".
  We thank the reviewer for this helpful remark. Indeed, the current structure may be misleading as it implies the models were part of GLOFRIM. This is, however, not the case and thus we will update the manuscript structure accordingly.
- Appendix A is not present.
  The appendix is provided in the assets of the manuscript under the header supplements. However, from the file name provided there it does not become clear that it is appendix A despite it was the original name of the uploaded file. We will try to fix this together with the Copernicus publishing team.
- Some of the authors of this paper are also authors of the PCR-GLOBWB model.  I have no correction to offer here, but this just feels a little odd.  Perhaps the paper might be stronger if an outside model had been used.
  We can guarantee that the fact that one author of this manuscript also worked on the PCR-GLOBWB model (and in fact another one on the CaMa-Flood model) has no influence on the set-up or results of the study presented here.
- D-Flow FM is referenced here and in Figure 1, but not discussed in the paper.
  Many thanks for pointing at this. Indeed, the model is not discussed in the paper as it was not used. A description can be found in the supplement of the paper as well as in the online resources. What is misleading is, that the associated header is 2.2.4 which is part of 2.2 "The models". This is not correct and 2.2.4 should stand individually as 2.3 which will improve clarity.
- What are the time steps of the models?
  We kindly thank the reviewer for mentioning this ambiguity. The time step of the hydrological model is daily and the hydrodynamic models (CaMa-Flood and LISFLOOD-FP) use an adaptive timestep scheme. The data between the models is exchanged at a daily time step. We will update the revised manuscript accordingly.
- Why are KGE and NSE chosen? One or two sentences on why these are the appropriate measures would be useful.

Many thanks for this critical remark, we will append the information to the revised manuscript. The metrics were chosen as they are very common in discharge analysis in general. Also, the KGE and its components provide information about a wide range of analytics. The NSE was chosen as it is particularly sensitive for deviations in peak discharge and thus powerful for flood hazard assessments such as shown in the submitted manuscript. We will add a sentence to the manuscript explaining this.

- Why is KGE > 0.7 significant?
Thank you for this comment. As with many hydrologic performance metrics, there is no clearly defined threshold when performance is deemed good or to have "skill". Since the maximum value of the KGE is 1.0, KGE values above 0.7 can, in our view, be interpreted as skillful as most uncalibrated discharge simulations typically range in lower spheres.

- What are KGE_α and KGE_β? I can guess what they are, but they should be explained.
Many thanks for reporting this inconsistency in the manuscript. They are indeed explained in section 3.1.2 but are named $\alpha$ and $\beta$ there, respectively. In the revised manuscript, we will make sure that the nomenclature is consistent throughout the document.

- I applaud the authors for making their code publically available. In the interest of open science, are the model runs available, as well? We thank the reviewer for those kind words. We were already discussing whether we want to make (parts of) the model input data to make the simulations as open and FAIR is possible.